# Multiplex Digital Spatial Profiling in Breast Cancer Research: State-of-the-Art Technologies and Applications across the Translational Science Spectrum

**DOI:** 10.3390/cancers16091615

**Published:** 2024-04-23

**Authors:** Matilde Rossi, Derek C. Radisky

**Affiliations:** Department of Cancer Biology, Mayo Clinic, Jacksonville, FL 32224, USA; rossi.matilde@mayo.edu

**Keywords:** spatial biology, breast cancer, single-cell analysis, spatial transcriptomics, spatial proteomics

## Abstract

**Simple Summary:**

Multiplex digital spatial profiling is a newly emerging approach for investigating multiple RNA transcripts and proteins simultaneously in the context of intact tissue. This review describes the different spatial technologies for the study of breast cancer, providing examples of their applicability in research and clinical settings to guide diagnosis and prognosis and to inform treatment decisions.

**Abstract:**

While RNA sequencing and multi-omic approaches have significantly advanced cancer diagnosis and treatment, their limitation in preserving critical spatial information has been a notable drawback. This spatial context is essential for understanding cellular interactions and tissue dynamics. Multiplex digital spatial profiling (MDSP) technologies overcome this limitation by enabling the simultaneous analysis of transcriptome and proteome data within the intact spatial architecture of tissues. In breast cancer research, MDSP has emerged as a promising tool, revealing complex biological questions related to disease evolution, identifying biomarkers, and discovering drug targets. This review highlights the potential of MDSP to revolutionize clinical applications, ranging from risk assessment and diagnostics to prognostics, patient monitoring, and the customization of treatment strategies, including clinical trial guidance. We discuss the major MDSP techniques, their applications in breast cancer research, and their integration in clinical practice, addressing both their potential and current limitations. Emphasizing the strategic use of MDSP in risk stratification for women with benign breast disease, we also highlight its transformative potential in reshaping the landscape of breast cancer research and treatment.

## 1. Introduction

Breast cancer now surpasses lung cancer as the most frequently diagnosed cancer worldwide, with 2.3 million individuals diagnosed in 2020 and projections suggesting an increase to 3.19 million by 2040 [1,2,3]. In the United States, it is estimated that 43,700 lives were claimed from breast cancer in 2023 [3,4], highlighting the urgent need for innovative diagnostic and treatment strategies. The rising incidence of breast cancer is partly due to ongoing demographic shifts and lifestyle changes, including delayed childbirth, the reduced frequency and duration of breastfeeding, unhealthy diets, and sedentary behavior, alongside advancements in detection methods [5], emphasizing the need for continued innovation in cancer care.

To reduce breast cancer mortality, identifying women at greatest risk is crucial. Risk factors for breast cancer vary amongst women and can be categorized into modifiable and non-modifiable factors. Non-modifiable factors include mutations in genes that confer increased risk (such as BRCA1 and BRCA2), race/ethnicity, breast tissue density, history of benign breast disease, and hormonal milestones such as age at the first menstrual period and age at menopause [5]. Modifiable factors include alcohol intake, smoking, dietary habits, and exposure to harmful drugs and chemicals [5,6]. The evolving understanding of how these diverse breast cancer risk factors interact has led to the development of sophisticated multifactorial risk-assessment models that integrate molecular data, such as gene expression profiles and genetic polymorphisms, and epidemiological data, such as lifestyle and behavioral patterns. These models significantly enhance the precision of individual breast cancer risk estimates, allowing for more targeted prevention strategies and early interventions.

Despite advancements in risk-assessment models, a significant gap persists in our ability to decipher the heterogeneity of breast cancer. Traditional diagnostic methods that homogenize tissue samples can lose critical spatial information that is essential for understanding the microenvironment of cancer cells—a key determinant in cancer progression and the response to treatment. By preserving this spatial context, multiplexed digital spatial profiling (MDSP) captures the heterogeneity within a tumor, revealing how different regions of the same tumor may respond differently to treatments. This precision not only improves our understanding of breast cancer initiation and progression but is also critical in personalizing the treatment of breast cancer patients. MDSP allows for a highly multiplexed, quantitative, and spatially resolved profiling of proteins and RNAs within a tissue sample, providing efficiency in extracting rich data from a single biopsy and minimizing the need for multiple invasive procedures, which reduces patient discomfort and healthcare costs. The data obtained from a single MDSP analysis can guide treatment decisions more effectively than multiple traditional tests, making it a valuable tool in both diagnosis and treatment planning. Its applicability is not restricted to malignant conditions; it can also offer insights into benign breast diseases for better risk stratification. By comparing the molecular profiles of benign and malignant tissues, MDSP can help identify potential early markers of malignant transformation, enabling the discovery of more effective prevention strategies. Furthermore, MDSP may be employed in developing personalized prevention strategies by correlating individual risk-factor profiles with specific molecular signatures. At present, MDSP is primarily used in the research setting; however, these methods are increasingly being applied in studies stemming from clinical trial data and, in some instances, are the method of choice to test exploratory endpoints in clinical trials. Such investigations represent the foundational studies that are building a body of evidence for its application to clinical practice.

In this review, we explore a range of MDSP platforms, each providing a distinct contribution to advancing breast cancer research. NanoString GeoMx stands out for its high-precision spatial profiling, while 10× Genomics’ Visium provides in-depth transcriptomic analysis. Imaging Mass Cytometry, Akoya PhenoCycler, and PhenoImager Fusion provide detailed molecular imaging and advanced phenotyping. These platforms collectively illuminate the complex molecular landscape of breast tumors. These insights have proven useful in identifying disease subtypes, refining prognostic predictions, and tailoring therapeutic strategies. Integration of MDSP with other diagnostic modalities, such as anatomic or functional imaging, can enhance both its sensitivity and specificity. By overlaying molecular and spatial data with clinical imaging, clinicians can obtain a more precise understanding of tumor heterogeneity, as has been accomplished in other clinical areas [7], leading to more personalized and effective treatment strategies. Additionally, the combination of MDSP with different omic approaches, such as genomics and metabolomics, can deepen our understanding of both the molecular drivers of breast cancer and identify novel therapeutic targets. Overall, MDSP represents a promising approach to enhancing our understanding of breast cancer, from risk prediction to therapeutic intervention. This review aims to elucidate the potential and promise of MDSP, providing a comprehensive view of its current applications and future perspectives in improving breast cancer care.

## 2. The Importance of the Tumor Microenvironment in Breast Cancer Research

Understanding the tumor microenvironment (TME) is critical for driving advancements in cancer research and treatment development. The TME is a complex and dynamic system that includes not only neoplastic cells but also supporting cells like fibroblasts, endothelial cells, adipocytes, and immune cells, as well as non-cellular components, including the extracellular matrix (ECM) and soluble factors such as chemokines, cytokines and growth factors [8]. The complex interplay between these components influences tumorigenesis, metastasis, and treatment responses in breast cancer. As the stromal components of the TME are tightly linked to metastatic processes, the structural properties of the TME are also linked to clinical outcomes, including chemoresistance and recurrence [9,10]. The advent of spatially-resolved high-plex molecular profiling technologies has revolutionized the way we characterize the breast cancer TME by allowing simultaneous protein and RNA profiling whilst maintaining the spatial context of these molecules, providing a more detailed and holistic view of the TME [11].

Key processes in cancer progression, such as the recruitment of metabolic resources, immune evasion, and epithelial-to-mesenchymal transition, occur within the TME, driven by the interactions of tumor, immune, and stromal cells through spatial networks [12]. Deciphering these spatial networks is critical, and MDSP technologies provide this information by quantifying and preserving the location of relevant markers. The value of spatial exploration in cancer biology—which is at the core of the innovation of MDSP technologies—is also reinforced by the understanding that a cell’s function and state are influenced by its spatial context and interactions with neighboring cells [13]. In fact, abnormal spatial organization of tissues is a histopathological feature that pathologists rely on to make clinical diagnoses [14]. Furthermore, a cell’s proximity to other cells and non-cellular structures informs its phenotype and state, and it determines which signals the cell receives, whether from cell–cell interactions or soluble signals exchanged between neighboring cells [15]. Thus, the ability of MDSP to map these interactions at the transcriptomic and proteomic levels within the tissue context offers new insights into the biological processes and pathological markers of the TME in breast cancer. Next, we describe the specific technologies that enable spatial profiling and their applications in breast cancer research.

## 3. MDSP Technologies

### 3.1. NanoString GeoMx in Breast Cancer Research

NanoString GeoMx represents a significant advancement in the spatial analysis of mRNA and proteins in tissue samples, addressing limitations of traditional in situ hybridization and immunohistochemistry, which typically analyze only one to four genes or markers at a time. NanoString GeoMx facilitates the high-throughput quantification of proteins and immune cells in both stromal and intraepithelial compartments in breast and other cancers, illustrating its broad utility in oncological research [16].

The GeoMx workflow involves tissue preparation and staining, imaging, region of interest (ROI) selection, the photocleaving of oligos and oligo collection, and digital quantification (Figure 1). The process begins with the incubation of formalin-fixed paraffin-embedded (FFPE) tissue sections with oligo-conjugated antibodies (for spatial proteomics) or oligo-conjugated RNA probes (for spatial transcriptomics), followed by imaging to select ROIs. The instrument contains two digital micromirror devices that then direct UV light to cleave oligos from the antibodies or RNA within the boundaries of the ROI. The released oligos are then collected using a microcapillary and transferred to a well plate for sequencing or digital counting with the nCounter system [17].

In recent studies, GeoMx has provided insightful findings in breast cancer research. Schlam et al. (2021) used it to profile the TME of primary and metastatic HER2+ breast cancer [18], discovering differences in immune cell infiltration and the expression of immune activation markers, with primary tumors showing enhanced immune cell infiltration within the stromal compartment compared with metastatic disease. Similarly, Omilian et al. (2021) examined immune infiltration differences between African American/Black and European American/White females in estrogen receptor (ER)-positive and ER-negative breast cancer, revealing demographic-specific disparities [19]. Moreover, Morrow et al.’s investigation into IL6/JAK/STAT3 signaling in triple-negative breast cancer (TNBC) linked high expression of STAT3 in tumor-associated stroma with adverse clinical outcomes, reduced CD4+ T-cell infiltration, and increased tumor budding. GeoMx spatial profiling demonstrated differential gene expression in high-STAT3 tumors, shedding light on TNBC heterogeneity and the intricate dynamics of the TME [20].

While GeoMx enables the use of a custom-defined set of biomarkers that includes approximately 18,000 transcripts and 100 validated protein targets, highlighting its capacity for broad molecular profiling, the CosMx spatial molecular imager complements this information by offering RNA and protein expression analysis at the single-cell and subcellular levels, with a focus on 1000 RNAs and 100 proteins [21]. This complementary suite of technologies, with GeoMx’s broad-scale spatial profiling and CosMx’s detailed single-cell analysis, provides a comprehensive toolkit for unraveling the complexities of the TME in breast cancer, from the macroscopic landscape down to the minutiae of cellular interactions. CosMx uses an automated microfluidic imaging system that gathers expression data from cyclical in situ hybridization chemistry [22]. RNA or antibodies are linked to a target binding domain or site-specific linker, respectively, with an in situ hybridization probe (containing a readout domain), a photocleavable linker, and a fluorescent reporter. After traditional target retrieval via permeabilization and fixation, reporter sets are added to the tissue, and the slide is assembled in a flow cell and CosMx machine. Following binding to the complementary sequence the fluorescent reporters are UV-cleaved and washed, enabling the next set of reporters to be added [23]. The readout domain on the in situ hybridization probe enables readout directly from the tissue without the need for the nCounter system. As such, CosMx is considered an image-based spatial transcriptomic approach, in contrast to GeoMx, which collects barcoded RNA and processes it for sequencing (sequencing-based approach) [24].

### 3.2. 10× Genomics’ Visium in Breast Cancer Research

Visium spatial gene expression by 10× Genomics provides a molecular profiling method to map the entire transcriptome within the tissue context. This technology enables the measurement of gene expression from a tissue sample and identifies where this activity occurs, revealing the complex interplay between cells in their native environment. The Visium process starts with tissue sectioning (using either freshly frozen or FFPE samples) and can involve either H&E staining for morphological insights or immunofluorescence for protein co-detection. Following imaging, the coverslip is removed to proceed with barcoding and library construction on glass slides coated with mRNA capture probes. For frozen samples, mRNA is released, captured by adjacent probes, and converted into barcoded cDNA for sequencing. In contrast, FFPE samples undergo probe hybridization to target genes, with ligated probes binding to capture probes after permeabilization. The sequencing libraries generated for both sample types create a spatially resolved transcriptome map (Figure 2) [25].

In contrast with NanoString GeoMx/CosMx, described above, Visium captures the entire coding transcriptome through a non-targeted approach, whereas the NanoString platforms employ panel-based approaches for targeted RNA capture. Visium’s specificity for polyadenylated RNA also differs from the capability of GeoMx/CosMx to capture specific RNA species with a custom-designed probe [26]. The strengths and weaknesses as well as assay variations between GeoMx and Visium have been objectively assessed specifically in breast cancer tissue by Wang et al. (2023) [27].

Recent studies highlight how Visium can complement histopathology to unravel the tumor microenvironment, advance biomarker discovery, and identify novel therapeutic targets [28]. For instance, Liu et al. (2023) profiled metastatic axillary lymph nodes paired to primary breast tumors to characterize early dissemination events, uncovering key metabolic changes during lymph node metastasis [29]. By combining spatial transcriptomics with single-cell RNA sequencing, they linked a metabolic switch between glycolysis to OXPHOS to early-disseminating breast cancer cells, noting these cells’ location at the tumor’s leading edge. Similarly, Bassiouni et al. (2023) explored spatial transcriptomic profiles of African American and Caucasian patients with TNBC, uncovering transcriptional substructures contributing to intratumoral heterogeneity and racial disparities in tumor hypoxia and immune infiltration [30]. Additionally, Foster et al. (2022) identified distinct cancer-associated fibroblast (CAF) clusters within mouse tumors, identifying mechanoresponsive, steady-state-like, and immunomodulatory subgroups based on spatial gene-expression profiles [31]. They found that these subtypes are located in spatially distinct regions of the tumor and that their distribution is disrupted by immune checkpoint inhibition. A reduction in steady-state-like superclusters and an increase in immunomodulatory superclusters were observed following treatment, suggesting the potential for CAF-targeting therapies to alter tumor growth by modulating the balance among these CAF subtypes.

### 3.3. Imaging Mass Cytometry (IMC) in Breast Cancer Research

IMC combines the analytic precision of mass cytometry with spatial resolution to provide high-resolution, multi-parametric imaging of cellular systems within tissues. Utilizing a cytometry by time-of-flight (CyTOF) approach, IMC enables the simultaneous analysis of multiple markers at the cellular level while preserving their spatial context within tissue samples [32]. The method involves staining archival FFPE or freshly frozen tissue samples with antibodies conjugated to unique metal ions. A pulsed laser linked to a mass spectrometer ablates pre-defined ROIs, turning them into particle plumes for CyTOF analysis, creating a detailed map of protein expression across the tissue landscape (Figure 3).

IMC has found many applications in oncology, notably in elucidating tumor biology and defining tumor cell and tumor microenvironment interactions that are crucial for understanding responses to immunotherapy [33,34,35,36,37]. In breast cancer research, IMC has shed light on the spatial distribution of CD8+ T-cells and the extracellular domain of HER2, providing insight into cytotoxic T-cell responses against HER2-positive breast cancer [38]. Further, it has facilitated the identification of distinct CAF populations in the breast cancer microenvironment that differentially influence tumor evolution. For instance, Cords et al. (2023) used IMC to corroborate scRNA-seq findings at the protein level and to examine the spatial distribution of CAFs, including their proximity to tumor borders and structures such as vessels and different cell neighbors, thereby revealing distinct CAF phenotypes [39].

Advancements in three-dimensional (3D) IMC have furthered our understanding of tissue structure and function, enabling a deeper understanding of cellular and microenvironmental heterogeneity and organization within breast cancer samples. Kuett et al. (2022) [40] applied 3D IMC to human breast cancer samples, unveiling cellular arrangements and microenvironmental complexities previously unappreciated in two-dimensional analyses [27]. These findings emphasize the potential of IMC to offer novel diagnostic and prognostic insights that enhance our understanding of breast cancer biology.

### 3.4. Akoya PhenoCycler and PhenoImager Fusion in Breast Cancer Research

Akoya’s PhenoCycler and PhenoImager Fusion transcend traditional immunohistochemistry (IHC) by using multiplex immunohistochemistry (mIHC) and multiplex immunofluorescence (mIF) for the concurrent detection of multiple protein markers in a single FFPE tissue section. The PhenoImager process involves deparaffinization, rehydration, antigen retrieval and blocking of the tissue, followed by incubation with primary and secondary antibodies. The secondary antibody, conjugated with horseradish peroxide, reacts with a fluorophore-conjugated tyramide. These oxidation reactions generate intermediates that bind near an epitope on the tissue through a covalent bond with tyrosine residues [41,42]. After stripping the antibody complex, the fluorophore remains covalently bound to or next to the epitope, allowing the process to be repeated for each marker. The cycle is repeated for each marker of interest, and the resulting slides are scanned by spectral imaging and analyzed using specialized software (e.g., InForm) that employs a reference library containing emission spectrums of the individual fluorophores and an autofluorescence slide to differentiate the fluorescent signals and eliminate autofluorescence, ensuring precise marker identification (Figure 4).

The PhenoCycler (previously CODEX) uses DNA-barcoded antibodies to detect over 40 biomarkers simultaneously. This system automates the labeling, imaging, and barcode-removal process across tissue sections stained with a comprehensive antibody panel, enabling multiplexed marker visualization (Figure 5) [41,43].

Griguolo et al. (2022) used mIF to dissect the immune microenvironment in breast cancer brain metastases, identifying subtype-dependent differences linked to overall survival [44]. The mIF approach enabled the authors to define the prognostic significance of immune subpopulations such as intra-tumoral infiltrated CD8+ and CD163+ cells and PD-1/PD-L1 spatial interactions, determined through spatial analysis. Such detailed immune-cell quantification offers insights into correlations between immune infiltration patterns and the overall survival and response to therapy [45], suggesting potential targets for immunotherapy. This characterization of the breast cancer microenvironment highlights the distinct molecular pathways implicated in cancer progression and potential immunotherapeutic strategies.

### 3.5. Other Notable Advanced Spatial Omics Technologies

Spatial profiling of proteins and mRNA can be achieved through imaging-based technologies or sequencing-based technologies, which have evolved since the early 1980’s [15,46]. So far, we have detailed the application and workflows of widely distributed MDSP technologies such as GeoMx and Visium, which, in 2020, were recognized as methods of the year by Nature Methods [22]. However, other notable methods in the milieu of spatial omics exist, as well as other emerging technologies, which are increasing the availability of MDSP. One example, multiplex error-robust fluorescence in situ hybridization (MERFISH), can resolve up to 10,000 gene targets [47,48]. Other methods, such as spatio-temporal enhanced resolution (Stereo-seq), have been developed for systemic applications, such as for the study of gestation embryos to detect transcriptome changes in mouse organogenesis [49]. In an effort to improve the mapping of gene expression at the single-cell level, Rodiques et al. (2019) developed Slide-seq, an NGS-based approach, which, in 2021, released its improved version, Slide-seqV2 [50,51]. Another innovative approach, deterministic barcoding in tissue for spatial omic sequencing (DBiT-seq), uses microfluidics to apply barcodes to the tissue [14,52].

## 4. Advantages and Limitations of MDSP Technologies

MDSP platforms like Akoya PhenoImager and PhenoCycler surpass traditional single-stain IHC methods in capability and versatility. The PhenoImager excels in high-multiplexing, allowing for multiple rounds of staining without disrupting the dye on the target protein by utilizing a reliable stripping process between cycles, while the PhenoCycler iteratively cycles through staining, imaging, and barcode removal for fluorescent antibodies, overcoming spectral bleed-through limitations with photocleavable barcodes for proteomic targets and spectral unmixing techniques [53]. IMC provides higher sensitivity and lower background noise by employing metal tags instead of traditional fluorophores. MDSP technologies are particularly advantageous in spatial transcriptomics, offering insights into transcriptional patterns within the tissue architecture that bulk and single-cell RNA sequencing cannot provide [54]. Furthermore, MDSP technologies like 10× Genomics’ Visium support exploratory analysis without the need for predefined targets, a significant step forward from the limitations of IHC and in situ hybridization [55]. Their integrated imaging and bioinformatics platforms allow for whole-slide or area-specific characterization, although this approach requires guidance from experienced pathologists for ROI selection to ensure analytical accuracy [17].

However, these techniques have limitations with respect to sample handling, reproducibility, quantification, and accuracy. Sample preparation can be a source of data variability, and the age of the FFPE blocks and sample fixation conditions should be considered to avoid intra- and inter-tissue variability [56]. Other tissue requirements include avoiding areas of necrosis and hemorrhage, limiting the tissue area for analysis [57]. The type of spatial technology assay can also have different stress effects on the tissue throughout the experiment; for example, the PhenoImager protocol exposes tissues to multiple heating cycles for antigen retrieval, which can compromise tissue integrity, and the repeated cycles under antigen-stripping conditions can potentially affect the integrity of some epitopes, necessitating strategic planning in the staining sequence [53]. Another source of variability that may influence the reproducibility of MDSP experiments is the lack of guidelines to determine the minimum number of regions required to identify the main cell phenotypes, as proposed by Bost et al. (2023), as well as size of the ROIs and the number of cells in each ROI [57,58]. Addressing differences in data distribution, such as signal intensities between samples, will require the development and broader utilization of standardized quality control and normalization techniques [59]. With regards to accuracy, spatially resolved transcriptomics depends on the availability of high RNA quality in the tissue specimen; improved RNA recovery methods have been proposed to improve the robustness of the methods [60]. Workflows for the analytical decomposition of cell type mixtures in methods that do not provide single-cell resolution would benefit from computational models to improve their sensitivity and resolution [61]. There are other method-specific limiting factors: NanoString CosMx, which offers single-cell transcriptomic analysis, is constrained by a predefined panel of probes, while IMC is not compatible with viewing stained tissue sections to assess staining inconsistencies or artifacts. IMC also suffers from variations in signal intensity, depending on the type of machine used and the time from staining [62]. Finally, resolution is a major distinguishing factor amongst MDSP platforms that strive to achieve this at the single-cell level. For instance, the Visium platform uses a relatively large probe diameter of 55 µm, capturing the signal from multiple cells and diluting the single-cell resolution analysis [54]. However, resolution continues to be improved, such as in the latest development of Visium HD, which features 2 × 2 µm^2^ barcoded squares to enable single-cell scale spatial resolution.

Despite these drawbacks, the benefits of MDSP technologies in revealing the complexity of the tissue microenvironments, elucidating cellular interactions, and informing targeted treatment strategies are considerable. The MDSP techniques discussed here, except for IMC, are non-destructive in nature, meaning that the samples can be preserved for long-term storage and utilized for further staining by H&E or IHC; slides can also be stripped and re-probed to enable re-staining with a different antibody panel of interest, as in the case of GeoMx [17]. These methodologies have shown particular promise in breast cancer research, unlocking new avenues for personalized treatment strategies by revealing intricate cell–cell dynamics.

Selecting the most appropriate MDSP method involves considering factors like cost, research phase, and the specific research question. For example, IMC is ideal for interrogating small areas of interest in tissues where sensitivity is needed, whereas GeoMx or Visium might be preferred for broader, high-throughput discovery phases. For targeted investigations, IMC or PhenoCycler can focus on specific markers, such as immune-related signatures. An overview of the advantages and limitations of each technology is presented in Table 1, facilitating informed decisions in the context of research objectives.

## 5. MDSP Integration with Breast Cancer Diagnostic Modalities

### 5.1. Enhancing Individualized Breast Cancer Early Detection, Diagnosis and Risk Prediction

MDSP technologies can significantly advance breast cancer diagnostics by complementing traditional screening methods and risk prediction models. Current models incorporate factors like hormonal, environmental, and genetic influences [5,63,64,65], yet the integration of MDSP offers deeper molecular insights that transcend conventional clinical metrics. This integration is poised to refine risk assessments and diagnostic precision. Despite the significant contribution of mammographic screening to reducing breast cancer mortality, its augmentation with multi-omics data—mirroring efforts in other medical fields such as traumatic brain injury, where NanoString GeoMx and magnetic resonance imaging (MRI) were combined [7,66]—remains relatively unexplored. However, the potential for MDSP to enhance the sensitivity, specificity, and resolution of standard imaging techniques like mammography and MRI is clear [67], providing a path to improved tumor detection and monitoring, crucial in breast cancer management.

The distinction between cancerous and benign tissue, traditionally requiring expert pathologists, presents a notable diagnostic challenge [13,68]. The application of MDSP in this context is highlighted in studies using NGS-based spatial transcriptomics, such as the pioneering work by Ståhl et al. in 2016 and subsequent improvements by Yoosuf et al. in 2020 [13,69]. They introduced a machine learning algorithm trained on pathologist-annotated regions and spatial transcriptomic data, achieving high accuracy in distinguishing non-malignant ductal carcinoma in situ and invasive ductal carcinoma regions [13,14,69]. The model was able to make this distinction with high accuracy even in regions not trained on. These approaches can be used to support pathologists in clinical decision-making by enhancing the differentiation of ductal carcinoma in situ from invasive ductal carcinoma in clinical biopsy samples.

The subtyping of breast cancer, traditionally based on classic IHC markers (ER, PR, and HER2) and pathological variables (tumor size, tumor grade, and nodal involvement) [70], is now enriched through insights from MDSP into tumor heterogeneity [71]. Carter et al.’s 2023 study using NanoString GeoMx in TNBC revealed that enrichment of CD40 and HLA-DR in the intraepithelial compartment is associated with better outcomes [16], demonstrating promise for this method in precision oncology. Similarly, the use of NanoString GeoMx by Stewart et al. (2020) [72] identified HLA-DR as a marker of disease relapse in TNBC, emphasizing the importance of distinguishing between intraepithelial and tumor tissue compartments for accurate disease prognostication [72].

IMC has also been used to identify correlations between cell phenotypes, cell–cell interactions, and patient prognosis in several subtypes of breast cancer. Ali et al. (2020) quantified 37 proteins in 483 tumors from the METABRIC cohort [73], while Jackson et al. (2020) demonstrated the superiority of single-cell pathology over traditional IHC in analyzing tumor tissue of 352 patients for 35 clinically established targets, including ER, PR, and HER2 [74]. Their findings revealed phenotypic clusters within specific regions or lesions, with the distribution pattern linked to overall survival. Vascularization and T-cell infiltration were associated with poorer outcomes, and regions with high T-cell and macrophage infiltration were associated with better outcomes. Onkar et al. (2023) used mIF to define how the spatial distribution of macrophage subsets and T-cells in invasive ductal and lobular carcinoma was linked to patient outcome [75]. These advancements underscore the potential of MDSP to enhance breast cancer diagnostics and risk stratification, paving the way for more effective and personalized treatment strategies.

### 5.2. Developing Tailored Prevention and Treatment Strategies

The current landscape of prevention strategies for high-risk women remains suboptimal, often resorting to drastic and irreversible procedures such as prophylactic mastectomy. Alternatively, women are recommended antiestrogen therapy, which has limited uptake due to concerns over side effects and the uncertain impacts on mortality reduction [76]. This situation underscores the need for novel treatments, particularly for BRCA1/BRCA2-mutation carriers, who lack chemopreventive treatment options. The work by Caputo et al. (2023) demonstrated the potential of spatial transcriptomics in this area [77]. Using NanoString GeoMx, they characterized the breast tissue epithelia and microenvironment in BRCA1/2-mutation carriers, revealing enhanced stromal-to-paracrine signaling and increased integrin receptor expression in stromal cells compared with controls. These insights pave the way for designing new prevention treatments for this high-risk group.

MDSP technologies can also play a crucial role in addressing challenges in breast cancer treatment resistance, overtreatment, residual disease, and recurrence. For example, McNamara and colleagues (2021) studied proteomic changes in HER2-positive breast cancer using NanoString GeoMx [78]. They found that spatial proteomic changes observed during HER2-targeted therapy could predict which tumors would achieve a pathological complete response, suggesting the utility of MDSP for guiding treatment escalation or de-escalation. Similarly, Lee et al. (2015) identified patterns of tumor-infiltrating lymphocytes that predicted pathological complete-response and longer disease-free survival correlations in TNBC patients receiving neoadjuvant chemotherapy, offering another biomarker set for therapeutic guidance [79]. Moreover, the insights from IMC have the potential to inform and refine treatment strategies for personalized treatment [80]. Wang et al. (2023) observed the effect of multicellular spatial organization on the response to immune checkpoint therapies, identifying spatial biomarkers as predictors of a therapeutic response [81]. Collectively, these studies emphasize the potential of MDSP in facilitating adaptive therapy [81], moving toward more individualized and effective cancer prevention and treatment paradigms.

MDSP technologies are also being used as a tool to identify patterns associated with therapy resistance. Use of 10× Genomics’ Visium has shed light on the role of pharmacogenes in drug distribution and efficacy. In the case of chemotherapy, drug concentrations vary across different regions of the tumor, and spatial transcriptomics has been shown to have the capability to capture spatially distinct pharmacogenes associated with chemoresistance [82]. Another study looking at biopsies during and post-neoadjuvant chemotherapy treatment identified the enrichment of chemokines and interleukins associated with a poor response to chemotherapy [83].

mIF-based assays are also showing promise in clinical trials and decision-making processes. Sanchez et al. (2021) advocate for incorporating mIF alongside standard clinical assays in clinical trials as their research identified pharmacodynamic biomarkers in the immuno-oncology of early-stage breast cancer [84]. A meta-analysis comparing the diagnostic accuracy of IHC, tumor mutation burden, gene expression profiling, and mIF in predicting a therapeutic response to anti-PD-1/PD-L1 therapy found that mIF showed superior AUC, positive predictive values, and positive likelihood ratios, suggesting an improved performance by mIF compared with traditional methods [85]. Furthermore, mIF is recommended for refining patient selection for immune checkpoint therapy, combining the assessment of PD-L1 status with tumor-infiltrating lymphocyte density for enhanced predictive value compared with standard IHC PD-L1 assays alone [86].

Thus, the incorporation of MDSP methodologies into clinical trials and treatment selection has the potential to advance personalized breast cancer care. MDSP facilitates patient stratification through unique molecular signatures, enabling a more tailored approach to treatment. It is particularly effective in distinguishing distinct molecular subtypes of breast tumors, guiding the selection of targeted treatments and immunotherapies. Moreover, MDSP can play a significant role in the real-time monitoring of treatment efficacy and the early detection of resistance, such as changes in immune-cell populations, protein expression, or the activation of alternative pathways. This allows for timely adjustments in therapeutic strategies, enhancing treatment outcomes. Although the application of MDSP is mostly still limited to laboratory research, its current inclusion in clinical trials for analysis or post-trial studies to correlate molecular features to patient outcomes is outlined in Table 2 and in exemplar studies by Radosevic-Robin et al. (2021) and Li et al. (2019) [83,87].

## 6. Future Directions

The application of MDSP to the management of benign breast disease (BBD) could be transformative for improving breast cancer risk prediction and designing personalized prevention methods. BBD, a known risk factor for cancer, could benefit from the ability of MDSP to identify subtle or spatially defined biomarkers that indicate the progression risk and prevention-treatment response. Biomarkers such as alterations in immune cells, extracellular matrix components, and protein expression, could help classify patients into distinct risk categories, improving the accuracy of predicting breast cancer onset and progression. Integrating MDSP with other diagnostic tools, including imaging and genomics, opens avenues for customized interventions ranging from lifestyle alterations to targeted therapies or preventive surgeries.

Implementing MDSP in clinical practice involves addressing key challenges to ensure its successful integration into breast cancer care. The standardization and validation of MDSP techniques are needed to avoid potential inaccuracies and to ensure consistent results from research labs to hospitals. It will be critical to develop clear protocols for tissue sample handling and MDSP analysis, with an emphasis on precision and reproducibility. This consistency is a prerequisite for the medical community to trust and adopt MDSP-driven insights and its integration into breast cancer care, bringing with it the promise of enhanced diagnosis, prognosis, and treatment.

Implementing MDSP in clinical practice also demands specialized expertise and training for healthcare professionals. Effective utilization of MDSP requires a deep understanding of the technology, its underlying principles, and the interpretation of its results. This may necessitate the development of educational programs and resources for healthcare professionals involved in breast cancer care, ranging from oncologists to pathologists. Moreover, the effective implementation of MDSP may require collaboration between multidisciplinary teams, such as oncologists, radiologists, pathologists, and bioinformaticians, to ensure a comprehensive understanding of the data generated and its implications for patient care.

Accessibility and cost considerations are also important factors to consider when implementing MDSP in clinical practice. The infrastructure required to perform MDSP, such as advanced imaging platforms and computational resources, may be beyond the reach of some healthcare facilities, particularly in resource-limited settings. Furthermore, the cost of MDSP technologies and reagents could be a barrier to their widespread adoption. As these technologies mature, the cost should decrease, becoming more broadly affordable, as has been the case for next-generation sequencing, which has become less expensive due to advancements in technology and market competition. The uptake of MDSP in healthcare could therefore mirror the evolution of next-generation sequencing, where usage in the clinical setting has increased as a function of decreasing cost [89]. To address these challenges, efforts must be made to develop more affordable MDSP technologies, and financial support may be necessary to facilitate the implementation of MDSP in healthcare settings with limited resources.

## 7. Conclusions

MDSP represents a significant advancement in breast cancer research, offering unparalleled insights into the tumor microenvironment. This technology enhances our understanding of disease progression and treatment responses, leading to improved risk prediction, diagnosis, and personalized treatment strategies. However, challenges in standardization, accessibility, and cost must be addressed for its broader clinical integration. Future directions include extending MDSP’s application to BBD and focusing on multidisciplinary collaboration and training for healthcare professionals. The integration of MDSP into clinical practice holds great promise for advancing personalized medicine in breast cancer care.

## Figures and Tables

**Figure 1 cancers-16-01615-f001:**
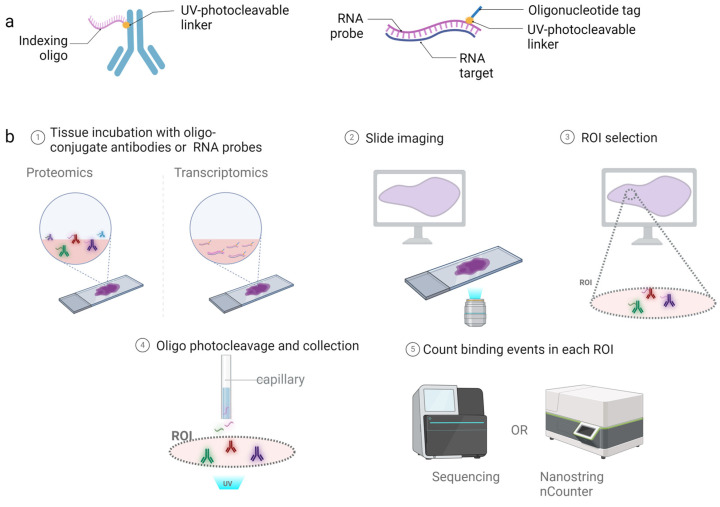
Schematic of NanoString GeoMx spatial transcriptomic and proteomic workflow. (**a**) Illustration of an oligo-conjugated antibody for protein detection or oligo-conjugated RNA probe bound to the target transcript. (**b**) The process begins with the conjugated probes being incubated with the tissue section, followed by imaging of the fluorescent markers for tissue location and selection of ROIs. Micromirror devices cleave the oligos within the boundaries of the ROI, and a microcapillary collects the cleaved oligos for nCounter or NGS readouts. Created with Biorender.com.

**Figure 2 cancers-16-01615-f002:**
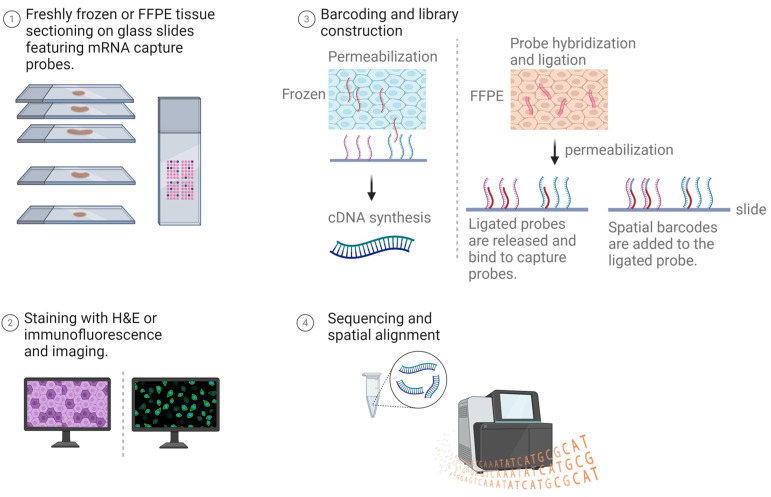
Workflow of 10× Genomics’ Visium for spatial transcriptomics. FFPE tissue or freshly frozen tissue are prepared on glass slides coated with mRNA capture probes. The slides are either stained with H&E or immunofluorescent markers and imaged. Following removal of the coverslip and tissue permeabilization, mRNA binds to probes for cDNA sequencing in frozen samples, or hybridized probes ligate to capture probes in FFPE samples for DNA sequencing. The resultant spatial transcriptome map overlays the reads onto tissue images. Created with Biorender.com.

**Figure 3 cancers-16-01615-f003:**
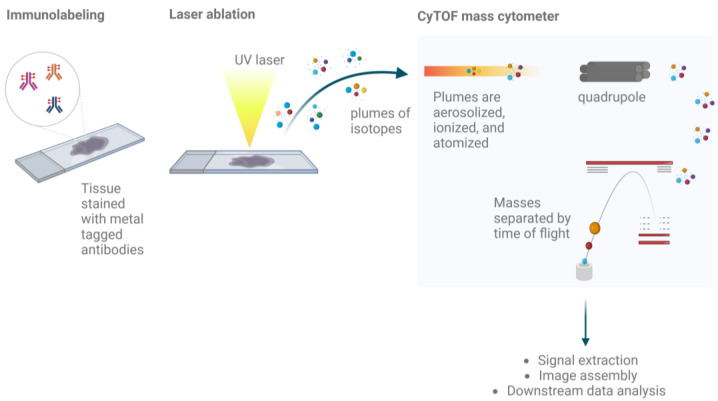
Workflow of IMC. Pre-processed FFPE or freshly frozen tissue are stained with a panel of antibodies, each of which are tagged with a metal ion that serves as a reporter. A UV laser ablates the tissue, generating plumes of isotopes that reach the connected mass cytometer. Ions are separated based on the mass-to-charge ratio as they pass through the quadrupole; the mass cytometer uses time-of-flight to identify the abundance of each metal reporter. Created with Biorender.com.

**Figure 4 cancers-16-01615-f004:**
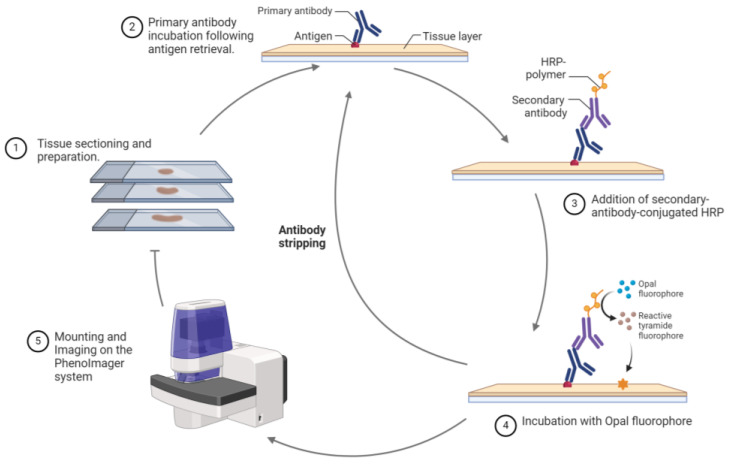
Workflow of the Akoya PhenoImager. FFPE tissue sections are prepared and stained using mIF procedures. After a heat treatment for antigen retreival, the tissue is blocked and incubated with the antibody of interest followed by incubation with a secondary antibody conjugated with HRP. The addition of the opal dye creates oxidized intermediates that covalently bind the tissue adjacent to the epitope. The primary and secondary antibody complex is stripped, and the cycle is repeated for the next marker of interest. The tissue is counterstained with DAPI for nuclei visualization. Created with Biorender.com.

**Figure 5 cancers-16-01615-f005:**
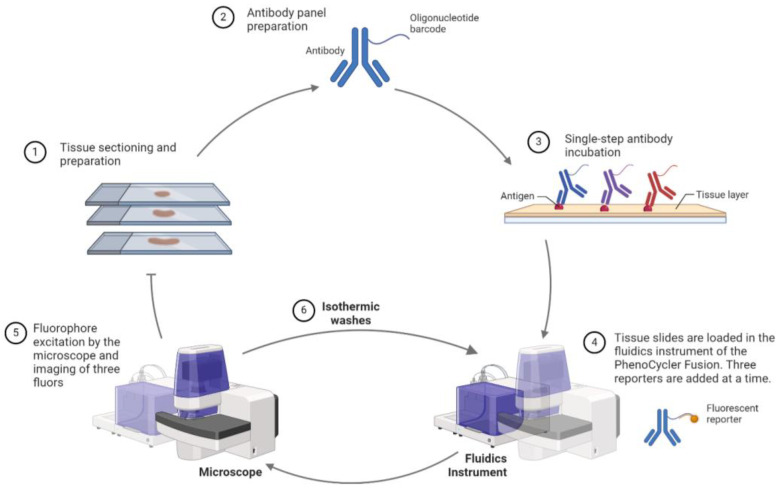
Workflow of the Akoya PhenoCycler Fusion. FFPE tissue sections are stained with antibody panels linked to oligonucleotide barcodes. The PhenoCycler cycles through staining, imaging, and barcode removal, enabling visualization of a comprehensive set of markers. Following antibody incubation, fluorescent reporters bind to the complementary barcode on the antibody. Three reporters are added at a time, and subsequent isothermic washes allow the addition of the following set of reporters. Created with Biorender.com.

**Table 1 cancers-16-01615-t001:** Summary of advantages and limitations of the MDSP technologies.

Method	Advantages	Limitations	Spatial Resolution	Compatible Sample Types
GeoMx	Utilizes photocleavable oligos, enabling flexibility in target selection High throughput for RNA and protein analysis User-defined regions for focused studies	Lacks single-cell resolution Limited to predefined biomarker panels, restricting spontaneous discovery	~10 µm	FFPE tissue block, fresh frozen tissue, tissue microarrays
CosMx	Enables single-cell and subcellular resolution Supports both RNA and protein analysis on whole slides Integrates seamlessly with GeoMx data for comprehensive profiling	Relies on pre-designed probe panels, limiting customization Higher cost due to single-cell resolution capabilities	Single-cell/subcellular	FFPE, fresh frozen, organoids, cultured cells
Visium	Barcode technology facilitates high-throughput analysis without fluorescent reporters Suitable for large-scale spatial profiling	Microslide size (55 µm) may blend signals from adjacent cells, complicating single-cell analysis	~55 µm	FFPE, fresh frozen tissue, tissue microarrays, PFA-fixed frozen tissue
IMC	Higher sensitivity and specificity with metal-ion labeling, eliminating autofluorescence Fixed antibody panel minimizes tissue degradation Ideal for detailed tissue composition studies	Limited number of detectible markers per slide (~40) Lower throughput due to extended imaging time and destructive nature of the method	1 µm	FFPE, fresh frozen tissue, tissue microarrays
PhenoCycler and PhenoImager	High multiplexing capability with biomarker co-expression Customizable antibody panels for tailored studies Improved sample stability and faster imaging times compared with other cyclic methods More affordable	Cyclic staining can lead to tissue degradation over multiple rounds Possible issues with spectral bleed-through despite advances in imaging techniques	~0.25 µm	FFPE, fresh frozen tissue, tumor microarrays

**Table 2 cancers-16-01615-t002:** Clinicaltrials.gov results from 9 February 2024.

NCT Number	Title	Status	Sponsor/Collaborator	MDSP Platform	Application
NCT03979508	Abemaciclib in Treating Patients With Surgically Resectable, Chemotherapy Resistant, Triple-Negative Breast Cancer	Recruiting	Mayo Clinic	NanoString GeoMx and Imaging Mass Cytometry	Evaluating the effects of abemaciclib on tumor-infiltrating immune cells
NCT04200768	FATLAS: Comprehensive Multi-level Characterization of Systemic and Mammary Adiposity in Breast Cancer Patients. (FATLAS)	Recruiting	Universitaire Ziekenhuizen KU Leuven	10× Genomics	Measuring upregulation or downregulation of pathways in adiposity and inflammation
NCT02977195 [88]	First in Human Evaluation of Safety, Pharmacokinetics, and Clinical Activity of a Monoclonal Antibody Targeting Netrin 1 in Patients With Advanced/Metastatic Solid Tumors (NP137)	Completed	Centre Leon Berard	10× Genomics, Visium	Confirming epithelial-to-mesenchymal transition gene expression changes

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
