# Peer review of "Multiplex Digital Spatial Profiling in Breast Cancer Research: State-of-the-Art Technologies and Applications across the Translational Science Spectrum"

_cancers, 2024, doi:10.3390/cancers16091615_

Round 1
Reviewer 1 Report
Comments and Suggestions for Authors
The article submitted by Matilde Rossi is a review of a case study of Multiplex Digital Spatial Profiling adapted to breast cancer. It is well organized and clear, focusing on technical aspects.
The limitations of PhenoImager from Line 316 are described, but it is important for users to define such technical issues; a purely technical introduction, not limited to breast cancer, should be included by introducing papers evaluating DMSP quantification and accuracy.
For example, the number of transcriptomics and target proteins that can be detected by GoeMX from Line 165 is described, but the quantitative quality of data such as reproducibility and qualitative characteristics such as what kind of proteins can be detected are not expressed, and the description of the measurement target is weak.
Evaluation of reproducibility should also be introduced. In addition, in order to improve the reproducibility, what should be taken care of in sample handling should be described; for PPPE samples, there is a problem that the reproducibility, such as the phosphorylation state of the protein, is extremely low depending on the sample handling. It would be beneficial to the reader if these issues were described with respect to MDSP.
Author Response
The article submitted by Matilde Rossi is a review of a case study of Multiplex Digital Spatial Profiling adapted to breast cancer. It is well organized and clear, focusing on technical aspects.
Response: We thank the reviewer for their supportive comments.
The limitations of PhenoImager from Line 316 are described, but it is important for users to define such technical issues; a purely technical introduction, not limited to breast cancer, should be included by introducing papers evaluating DMSP quantification and accuracy.
For example, the number of transcriptomics and target proteins that can be detected by GoeMX from Line 165 is described, but the quantitative quality of data such as reproducibility and qualitative characteristics such as what kind of proteins can be detected are not expressed, and the description of the measurement target is weak.
Evaluation of reproducibility should also be introduced. In addition, in order to improve the reproducibility, what should be taken care of in sample handling should be described; for PPPE samples, there is a problem that the reproducibility, such as the phosphorylation state of the protein, is extremely low depending on the sample handling. It would be beneficial to the reader if these issues were described with respect to MDSP.
Response to all comments: We appreciate the opportunity to improve the work with additional text on technical reproduction and sample preparation, which we have provided in lines 342-371 of the revised manuscript.
Reviewer 2 Report
Comments and Suggestions for Authors
The manuscript explores the potential of Multiplexed Digital Spatial Profiling (MDSP) technologies in breast cancer research and clinical practice. It discusses various MDSP platforms including NanoString GeoMx, 10x Genomics Visium, Imaging Mass Cytometry (IMC), and Akoya PhenoCycler and PhenoImager, and highlighting their capabilities in capturing spatially resolved molecular information within tissue samples. MDSP facilitates the characterization of the tumor microenvironment. By analyzing spatially molecular data, MDSP enables the identification of distinct tumor subtypes and subtle biomarkers associated with disease progression and treatment response, guiding personalized prevention and therapeutic strategies. Additionally, the integration of MDSP with traditional diagnostic modalities, such as imaging and genomics, enhances early detection, diagnosis, and risk prediction of breast cancer. Furthermore, the manuscript discussed the challenges and future directions for broader clinical integration of MDSP. Overall, MDSP holds great promise for advancing personalized medicine in breast cancer care.
Author Response
We thank the reviewer for their evaluation and supportive comments.
Reviewer 3 Report
Comments and Suggestions for Authors
1. In some methodological descriptions, the narrative might be a bit brief (though the images are very intuitive, which is good). For example, in reference [17], "Digital micromirror devices then guide UV light to cleave oligos from the antibodies or RNA within the boundaries of the ROI." It seems that some details may have been omitted since microscopes cannot actively drive UV light.
2. One criticism of MDSP is that the information it provides for diagnosis does not justify its cost. Taking 10x Genomic as an example, the cost per sample exceeds 5000 USD. However, it doesn't offer more diagnostic information than some cheaper sequencing diagnostic options (for instance, in breast cancer, using chips similar to the PAM 50 principle like Oncotype Dx versus using 10x Genomic). Should the article consider discussing this aspect of pricing and diagnostic efficacy? Otherwise, this article merely becomes an advertisement for MDSP technology.
Comments on the Quality of English LanguageFine
Author Response
1. In some methodological descriptions, the narrative might be a bit brief (though the images are very intuitive, which is good). For example, in reference [17], "Digital micromirror devices then guide UV light to cleave oligos from the antibodies or RNA within the boundaries of the ROI." It seems that some details may have been omitted since microscopes cannot actively drive UV light.
Response: We thank the reviewer for their careful attention; we have clarified that the micromirror devices and not the microscope directs the UV light in lines 140-141 of the revised manuscript.
2. One criticism of MDSP is that the information it provides for diagnosis does not justify its cost. Taking 10x Genomic as an example, the cost per sample exceeds 5000 USD. However, it doesn't offer more diagnostic information than some cheaper sequencing diagnostic options (for instance, in breast cancer, using chips similar to the PAM 50 principle like Oncotype Dx versus using 10x Genomic). Should the article consider discussing this aspect of pricing and diagnostic efficacy? Otherwise, this article merely becomes an advertisement for MDSP technology.
Response: This is an important point, which we have addressed in lines 532-536 of the revised manuscript.
Reviewer 4 Report
Comments and Suggestions for Authors
This review summarized the major MDSP techniques, their applications in breast cancer research, and both their potential and current limitations. It provides good guidance for both breast cancer mechanisms research and clinical diagnosis and treatment. But some issues need to be addressed before it can be further considered.
1. In line 24-26, there are two “we also address the challenges of …”, which should be revised.
2. The section 3 “MDSP technologies” should include some more advanced spatial omics technologies such as DBiT-seq, merFISH and so on.
3. In section 3.1 “NanoString GeoMx in Breast Cancer Research”, the workflow of CosMx technology should be explained in detail.
4. In section 4 “Advantages and limitations of MDSP technologies”, the authors mention that the limitations of the Visium platform are that it is difficult to achieve single-cell resolution, and signals from multiple cells are captured. However, the Visium platform has been upgraded and the resolution has now been greatly improved to almost single-cell resolution. At the same time, methods similar to Stereo-seq can also approach single-cell resolution. The recently published Slide-tag truly achieves spatial omics with single-cell resolution. Therefore, resolution does not seem to be a limitation of spatial omics.
5. In section 4 “Advantages and limitations of MDSP technologies”, line 327, the authors wrote: “Their generally non-destructive nature (with the exception of IMC) allows for long-term sample preservation for subsequent analyses.”
Please explain the definition non-destructive nature. In my opinion, most MDSP technologies are destructive, at least samples processed by 10X Genomics Visium cannot be analyzed repeatedly.
6. In Table 1, in the comparison of various methods, it is suggested to add a category: compatible sample types. As different technologies are limited in the types of samples they can process, this is crucial for clinical sample research, helping researchers to select suitable techniques for their studies.
Author Response
This review summarized the major MDSP techniques, their applications in breast cancer research, and both their potential and current limitations. It provides good guidance for both breast cancer mechanisms research and clinical diagnosis and treatment. But some issues need to be addressed before it can be further considered.
Response: We thank the reviewer for their supportive comments.
- In line 24-26, there are two “we also address the challenges of …”, which should be revised.
Response: We thank the reviewer for their careful attention. This has been revised, in lines 22-26 in revised manuscript.
- The section 3 “MDSP technologies” should include some more advanced spatial omics technologies such as DBiT-seq, merFISH and so on.
Response: We appreciate the opportunity to improve our work and have provided these details and references in lines 309-324 in the revised manuscript.
- In section 3.1 “NanoString GeoMx in Breast Cancer Research”, the workflow of CosMx technology should be explained in detail.
Response: We were pleased to provide these additional details in lines 172-183 of the revised manuscript.
- In section 4 “Advantages and limitations of MDSP technologies”, the authors mention that the limitations of the Visium platform are that it is difficult to achieve single-cell resolution, and signals from multiple cells are captured. However, the Visium platform has been upgraded and the resolution has now been greatly improved to almost single-cell resolution. At the same time, methods similar to Stereo-seq can also approach single-cell resolution. The recently published Slide-tag truly achieves spatial omics with single-cell resolution. Therefore, resolution does not seem to be a limitation of spatial omics.
- In section 4 “Advantages and limitations of MDSP technologies”, line 327, the authors wrote: “Their generally non-destructive nature (with the exception of IMC) allows for long-term sample preservation for subsequent analyses.” Please explain the definition non-destructive nature. In my opinion, most MDSP technologies are destructive, at least samples processed by 10X Genomics Visium cannot be analyzed repeatedly.
Response (to points 4 and 5): We have added extensive new text and references to address these points in lines 342-379 of the revised manuscript.
- In Table 1, in the comparison of various methods, it is suggested to add a category: compatible sample types. As different technologies are limited in the types of samples they can process, this is crucial for clinical sample research, helping researchers to select suitable techniques for their studies.
Response: We thank the reviewer for this suggestion; this information has been added to Table 1.
Round 2
Reviewer 3 Report
Comments and Suggestions for Authors
The suggestions provided earlier have been improved in this revision.